# A Text-Mining Approach to Compare Impacts and Benefits of Nature-Based Solutions in Europe

**Leydy Alejandra Castellanos [1,*] , Pierre-Antoine Versini [1], Olivier Bonin [2] and Ioulia Tchiguirinskaia [1]**

[1] HM&Co Lab, Ecole des Ponts ParisTech, 77455 Champs sur Marne, France;
pierre-antoine.versini@enpc.fr (P.-A.V.); ioulia.tchiguirinskaia@enpc.fr (I.T.)

[2] LVMT, Ecole des Ponts ParisTech, Université Gustave Eiffel, 77455 Champs sur Marne, France;
olivier.bonin@univ-eiffel.fr

\* Correspondence: leydy.castellanos@enpc.fr

**Abstract:** Worldwide, a large set of initiatives have been carried out aiming to understand the benefits offered by Nature-Based Solutions (NBS) in urban areas. The European Commission (EC) has founded different projects that have performed scientific literature reviews regarding this topic. To objectively compare their results and consolidate the consensus about the impacts and benefits of NBS, we performed a text mining analysis. This methodology coupled with a visual representation of the data allowed to convert the EC funding projects reports (corpus) into a meaningful structured analysis. This method demonstrated that despite the different literature review methodologies of each report, there are common trends exhibited by the results, e.g., the NBS installation as a strategy of urban resilience, the recognition of ecosystem services (ESS) delivered by nature in urban spaces, or the importance of the EC's supporting role in the promotion of NBS. In addition, some network specific trends have also emerged and complemented the analysis: the assessment of the NBS performance with indicators, the participatory planning approach to NBS (involving citizen and local communities) and the economic value of their services.

**Keywords:** text-mining analysis; European project; nature-based solutions; urban resilience

## 1. Introduction

The challenges induced by urbanization and climate change encourage the integration of green and blue infrastructures in the functioning of our cities. This aims to mitigate the intensity of extreme events such as floods and heat waves, to moderate the hot spots of dense urban areas (e.g., Urban Heat Island, UHI), and overall to improve urban resilience capacity. Several concepts have emerged to refer to these infrastructures promoting nature reintegration into build environment [1], such as Ecosystem-Based Adaptation (EbA), Sustainable Urban Drainage Systems (SuDS), Urban Green Infrastructure (UGI) or Nature-Based Solutions (NBS).

Aiming to build urban resilience, the European Commission (EC) introduced the NBS concept in its Horizon 2020 Program as "living solutions inspired and continuously supported by nature, which are designed to address various societal challenges in a resource-efficient and adaptable manner and to provide simultaneously economic, social, and environmental benefits" [2]. Moreover, the EbA concept has a climate change adaptation focus and aims towards the management and use of biodiversity and ecosystem services (ESS) as part of an overall adaptation strategy, reducing vulnerability and building resilience [3]. Besides, SuDs were conceived to reduce the impact on surface water drainage systems of urban flooding, such as rainwater runoff or sewer overflows. Meanwhile, the UGI concept is defined as a hybrid infrastructure of green spaces and built systems [4], which can contribute to

ecosystem resilience and human benefits through ESS. The EIT Climate-KIC Blue Green Dream project (https://hmco.enpc.fr/portfolio-archive/blue-green-dream/) appears as a pioneering European project devoted to this topic [5]. It first valued the role of NBS in the holistic management of urban green asset and water resources. Afterwards, this approach has been expanded to exploit the full range of ESS that NBS provide to society, yielding the Blue Green Solutions (BGS, [6]).

During the last decade, the research community has studied NBS worldwide to provide the impacts and co-benefits of these structures within cities, generating a huge amount of literature regarding this topic. This is the case of regulating services like storm-water management and temperature reduction. For instance, an important reduction of the peak flow in the Avola (southern Italy) stormwater sewer system was demonstrated by the implementation of green roofs and permeable pavements [7]. Further, the implementation of different NBS facilities—such as green roofs, rain gardens, green spaces, green swales, permeable pavements and retention basins—proved the minimization of both peak discharge and runoff volumes by up to 90% on the full range of rainfall event return periods at the urban project scale [8].

Regarding UHI mitigation, a literature review indicated that thanks to urban vegetation, there was a reduction of urban temperatures ranging from 0.5 to 4.0 °C, depending mainly on the green area and wind dynamics [9]. Similarly, a reduction of 1°C in land surface temperature during a heatwave event was performed by a 10% increase in vegetation cover in Melbourne (Australia) [10]. Meanwhile, further studies have targeted the analysis of "park cool island" to prove the cooling effect of urban park vegetation [11,12]. Other services provided by NBS, such as noise reduction [1,13], biodiversity restoration [14,15] and well-being [16], have been analyzed and quantified less.

Since these solutions are of interest to tackle the urban sustainability challenges of European cities, the EC has funded several projects to develop frameworks or guidelines for NBS implementation. In order to improve the base knowledge of physical, social and economic impact, extensive literature reviews have been carried out. However, it is quite difficult to compare the results of these literature reviews in order to consolidate the analysis and to find a consensus about the actual outcomes and benefits of all the proposed NBS. In this context, the aim of this research is to objectively compare the results of three Horizon 2020 (H2020) NBS projects by exploring their latest deliverables through a text-mining technique, and analyze them with a classical literature review. These three projects were chosen since they were pioneers in mapping NBS at the European scale and understanding their efficiency in tackling urban challenges. Consequently, these compose a reference widely cited by the scientific community and used by local governments as urban adaptation guides.

The purpose of text mining is to extract meaningful structured information from text data. Thus, text mining enables us to identify the key concepts and the main stakeholders described in the corpus, as well as their relationships. Text mining goes beyond lexical analysis by identifying patterns and attributes. Recently, text-mining tools were used to investigate and characterize how climate risk management issues are represented in 12 online strategic documents released by public authorities in Paris [17]. The applied method allowed to expose results not revealed through a qualitative analysis of the document, such as the integrated approach of public authorities to manage urban risk and the advocacy of NBS. Similarly, this approach was applied to understand the conceptual difference between urban resilience and urban sustainability, as well as the multi-level perspective for urban resilience and regional residence on peer-review literature of the Scopus database [18].

This paper is structured as follows. Firstly, the text data are presented, and their approaches and methodologies are identified (Section 2.1). Then, for each document, a mapping of the text data allows to extract the list of key terms used to create a graph. This graph exposes the main statistical properties of each text (Sections 2.2 and 2.3). Finally, the results of each document are analyzed individually (Section 3), and then compared to each other and to the scientific literature (Section 4), enabling to identify common and different approaches.

## 2. Methodology

### 2.1. Corpus

In a general way, to proceed with text mining, the first step is the collection of texts to be analyzed, known in linguistics as "text corpus". The corpus used in this study corresponds to three full-text reports. These reports are the final deliverables of the EC projects published between 2016 and 2017 about NBS impacts in urban areas at the European scale (Table 1): EKLIPSE, MAES-Urban Ecosystems and NATURVATION. These projects are presented bellow with some details summarized in Table A1.

The EKLIPSE Project was launched under the Horizon 2020 framework and aimed to build a sustainable and innovative EC support mechanism for evidence-based policy on biodiversity and ESS (http://www.eklipse-mechanism.eu/). The EKLIPSE Project was selected by the EC to establish a sustainable framework of design, development, implementation and assessment of NBS demonstration projects in the urban context using a holistic approach. The methodology was based on the study of scientific literature conducted either in research publications (320 peer-review articles or books) or in unpublished "grey" literature provided by organizational websites and web search engines (1223 units). Finally, additional references were added by the EKLIPSE Working Group (EWG) members and coordinators of NBS European Union (EU) projects (247 articles, books and reports). The resulting framework suggests 10 challenges that would support climate resilience at different spatial and temporal scales. These challenges were selected from the expert report on NBS supported by DG Research and Innovation-EC [19] and a recent review of the NBS frameworks [20]. Each challenge and the potential actions to mitigate, manage or promote it by means of UGI are included in the final report entitled "An impact evaluation framework to support planning and evaluation of nature-based solutions projects." The expected impacts and the effectiveness indicators based on the results of the literature review are included in this report; for this reason, this document constitutes the first text corpus for this study.

The Mapping Assessment of Ecosystem and their Services (MAES) initiative was launched by the EC in response to Action 5 of the EU Biodiversity Strategy to 2020 (https://biodiversity.europa.eu/maes). It aimed to define actions and plans to mitigate and stop biodiversity loss in EU. To achieve this goal, the ecosystems condition and their services were assessed and mapped. The 4th report of MAES initiative published in 2016 was exclusively dedicated to urban ecosystems and, for this reason, it constitutes the second corpus for analysis in this study. In this report, a survey and review of 54 scientific articles was used as a source of information to map urban green spaces, assess their condition and measure the ESS they provide. Additionally, ten case studies in Europe provided the base for real applications of NBS, allowing for a wide variety of ecosystem conditions to set the service assessment indicators. The results of the MAES initiative were used in the ESMERALDA project (Enhancing ecoSysteM sERvices mApping for poLicy and Decision mAking) to develop a 'flexible methodology' for pan-European, national and regional ESS mapping and assessment (http://www.esmeralda-project.eu/).

The technical feasibility of NBS implementation, as well as the assessment of their benefits to tackle urban challenges, have been widely analyzed by the academic community. However, despite the fact that NBS provide benefits and values for different stakeholders [21], there are very few studies on their economic valorization, such as cost–benefit analyses [16]. Hence, the NATure-based URban innoVATION (NATURVATION) project was launched to develop a comprehensive NBS assessment tool for urban areas with their economic implications (https://naturvation.eu/about). For this purpose, a literature review of 105 studies from the academic fields related to the financial and economic values of NBS was carried out, linking a monetary value to ESS (ecological and societal effect) and urban challenges. Because of the economic character of this report, it was considered as the third corpus for the text mining.

All the reports have commonly aimed to perform a literature review focused on current scientific knowledge of impacts, benefits and trade-offs provided by the implementation of NBS. However, each one adopted a specific methodology depending on its main objective and the stakeholders'

involvement in the project. The complementarity of these three reports (urban challenges and NBS assessment through indicators (EKLIPSE); mapping of key urban ESS through successful experiences supported by public authorities (MAES) and the economic and financial review of NBS (NATURVATION)) suggests to study the ensemble of approaches and stakeholders that could influence the impacts and benefits of NBS deployment. Such ensemble has not been yet identified in a classic qualitative literature review.

Given that every report is the result of a literature review of numerous articles, each report constitutes a text corpus that could first be explored individually.

**Table 1.** The three corpora for text-mining analysis: Nature-Based Solutions (NBS)-oriented projects.

| Report | Literature | Source |
|---|---|---|
| EKLIPSE—An impact evaluation framework to support planning and evaluation of nature-based solutions projects. Publication: 2017. | 320 peer-review scientific articles or books. | Database searches. |
| | 1223 pieces of grey literature. | Organizational website and web search engines. |
| | 247 articles, books and reports. | EKLIPSE Working Group. |
| Mapping and Assessment of Ecosystems and their Services (MAES) Urban ecosystems 4th Report. Publication: May 2016. | 54 scientific articles. | Science Direct database research. |
| | 10 Case studies. | Proposed by the research community. |
| NATURVATION—Review of Economic Valuation of Nature Based Solutions in Urban Areas. Publication: May 2017. | 105 studies from the academic work. | Database searches (meta-analysis). |

## 2.2. Text-Mining Analysis

Usually, literature review documentation has an extent which easily allows its reading and synthesis. However, while doing so, each reader would be biased by their own field of expertise. Indeed, even if the documents are easily readable, this research allows to analyze and compare them independently from the author's scientific domain of research.

To evaluate the content of each document, the text-mining analysis was carried out through Gargantex Blue Jasmine Version [22], an open-source software developed by the CNRS Complex Systems Institute of Paris Île-de-France (ISC-PIF) and its partners. The main purpose of Gargantex is to analyze the publication information of a large set of scientific literature articles (up to 1000). In this research, the publication information of articles (i.e., keywords, abstract, title, date of publication and author) is replaced by smaller units from the corpus texts, generally from the length of a paragraph (500 words approximately).

The Gargantex algorithm extracts automatically a list of words (or terms) from each corpus, based on their co-occurrence taking into consideration the integrality of the text. Each term is characterized by a status, according to three possibilities: map, stop and not flagged. The "map-list" contains the key or meaningful terms with the highest occurrences, which are used in the data visualization (Figure 1). The "stop-list" regroups the terms with few occurrences or without a meaningful conceptual contribution in the text (e.g., lists, tables or figures). Finally, the "not flagged" words are not classified in any of the previous lists (e.g., concentration, percentage or projects).

In order to refine the term-list, a terms-status modification and a lexicographic analysis is done by the user. This means keeping relevant terms with high occurrence that refer to NBS, environmental impacts, NBS governance and/or stakeholders (e.g., "blue spaces", "air pollutants", "local communities"). Likewise, the user can remove the meaningless words marked as "map", such as the author's name (e.g., "Baro", "Cohen", "Kabish", etc.) or nouns themselves (e.g., "tables", "units", etc.) Furthermore, synonyms and acronyms can be aggregated (e.g., "assessment"-"valuation" or "report"-"document"), as well as singular and plural forms (e.g., urban green infrastructure, green infrastructure or ugi), creating groups of terms and increasing their occurrences and co-occurrences.

Statistical properties of the corpus texts, such as their length, will impact the output statistical text-mining analysis. Length can be expressed in term of pages or paragraphs. Table 2 displays the statistical characteristics of the three corpus texts.

**Table 2.** Statistical feature of corpus text.

| Length | EKLIPSE | MAES | NATURVATION |
| --- | --- | --- | --- |
| Pages | 82 | 94 | 34 |
| Paragraphs | 143 | 254 | 58 |

### 2.3. Visual Data Representation and Quantitative Analysis

To facilitate the understanding of the text-mining results, the map-list terms are used to elaborate a visual representation in a graph or network form. A graph is a mathematical representation of interconnections between different elements and is composed of nodes (map-list terms) connected by edges (connections between terms). The networks are built with respect to the conditional proximity, an absolute measure that reflects the highest co-occurrences between two key terms in the same textual context [18]. Gargantex uses the Louvain Modularity Cluster Algorithm to create communities of nodes in the network (known as clusters), regrouping the nodes strongly related with the same color on the map. Each cluster is supposed to represent one approach, concept or idea conveyed by the text. According to the node topics, a name is allocated to each cluster by the user.

In order to characterize the properties of the network and the related clusters, a Gargantex GEXF file is extracted to be used as input data in the open-access interactive software of graph visualization and network analysis Gephi (URL: https://gephi.org/). By means of Gephi, all the statistic properties (attributes) of the network can be viewed and analyzed by the user. The attributes analyzed in this study are the node degree and the edge weight. They correspond respectively to the number of edges connected to a node and the strength of link between nodes (probability of co-occurrence between a pair of nodes). The node degree is a measure of centrality, which determines the influence or importance of a node in the network. Therefore, the centrality concept depends on the studied context. In the case of NBS, it might refer to the most significant NBS benefit or the most valuable solution in a complex urban context.

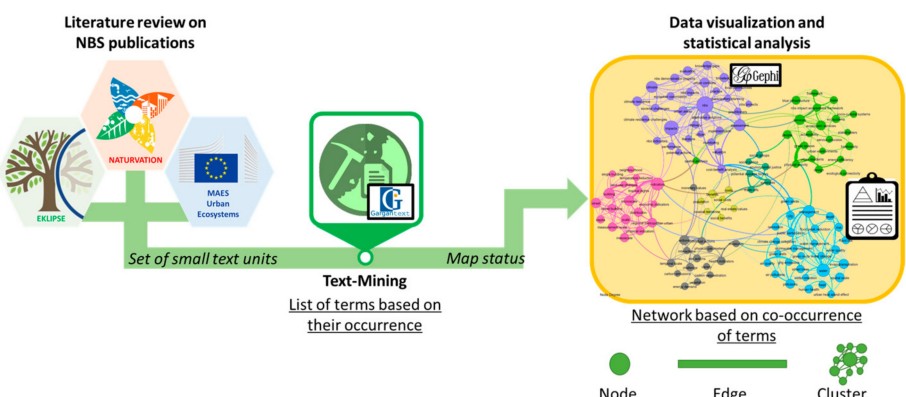

**Figure 1.** Text-mining and data visualization process.

The size node and the edge thickness in the networks indicate the node degree and the edge weight, respectively. Note that only nodes characterized by a degree higher than 3 are conserved in the analysis. Finally, the Gephi topological tools are used to improve the readability of the network by saving its original attributes (node degree and edge weight).

## 3. Results

In this section, the network representations of each corpus text are analyzed individually. The clusters composing the network are discretized and analyzed relying on the number of nodes integrating each cluster and their accumulated degree (sum of nodes degree). Furthermore, the resulting network attributes (node degree and edge weight) are examined deeply. This allows to draw conclusions or statements, which were supported by the scientific publication review.

### 3.1. EKLIPSE

The most important nodes in the EKLIPSE network were identified visually (Figure 2) and according to their degree (Table A2). They include "nbs" with the highest degree (36), followed by "impacts" (29) and "indicators" (22). This node degree classification suggests how NBS appears in the urban context, as a strategy to face environmental impacts and urban challenges. Then, "water" (22) appears as the environmental resource most impacted by "climate change" (19), compared with "air" (10) and "soil" (not even mentioned). "Management" and "assessment" are both positioned as keywords (with a node degree of 19), highlighting the possibility to gain services for the population. With a lower but nonnegligible degree (16 to 14) appear the terms related to scales like "building", "street" or "microscale", illustrating the importance of knowing well the NBS' scale of implementation to assess their performance in the urban planning process.

The NBS implementation evaluation framework in the EKLIPSE project is structured through seven clusters (Figure 2 and Table 3). Due to the quantity of nodes and their degrees, two clusters are particularly significant. We have named them "*NBS planning, governance and stakeholders*" and "*NBS to tackle urban challenges*".

The first of these clusters includes nodes like "nbs", "impacts", "climate", "assessment" and "urban context" characterized by the highest degrees (Table A3). Following the connections in this cluster, it is possible to identify: "nbs"-"urban areas", "nbs"-"societal challenges", "nbs"-"climate", "nbs"-"impacts", "nbs"-"participatory planning", "climate"-"societal challenges", etc. The nodes and connections show how the promotion of the NBS in urban areas through urban planning tools becomes a climate resilience strategy to create urban sustainability, as argued in recent literature [16,18,23]. This cluster also stresses the strong support of EC to deploy NBS in cities and its ambition to position Europe as a world leader in responsible innovation related to NBS [24]. Moreover, the inclusive approach and the involvement of local communities in the urban planning process known as collaborative planning [23] are represented by the nodes called "participatory planning", "local communities" or "citizen". Indeed, it has been noticed that NBS recognition by means of public participation ensures NBS viability and maintenance [20].

**Table 3.** Clusters of the EKLIPSE project, number of nodes and accumulated degrees.

| Cluster | Color | Number of Nodes | Accumulated Degree |
|---|---|---|---|
| NBS planning, governance and stakeholders | Purple | 34 | 311 |
| NBS to tackle urban challenges | Pink | 30 | 207 |
| NBS indicators and scale | Blue | 18 | 175 |
| Ecosystem services provided by the NBS | Green | 30 | 171 |
| Action, temporal scale and health benefits | Grey | 15 | 88 |
| Social and economic benefit of NBS | Yellow | 11 | 42 |
| Social opportunities | Dark green | 7 | 26 |

The "*NBS to tackle urban challenges*" cluster is the second most relevant cluster. It stands out in water management terms (e.g., "flood peak reduction", "flood risk", "water quality", etc.), emphasizing this regulation service provided by NBS. Furthermore, different types of UGI (e.g., "green roofs", "green walls", "street trees") are part of this cluster, stressing their capacity to attenuate the impacts produced by the urbanization process and climate change, improving the resilience capacity of urban

spaces [25]. Last but not least, several additional environmental urban challenges are noted in the cluster, such as "ghg emissions", "air pollutants" and "UHI effect".

The need to quantify the NBS' impacts and their efficiency has led to proposing some indicators. Hence, the "*NBS indicators*" cluster highlights this feature, taking into consideration the spatial scale where the solution is implemented, the impact to mitigate and the ESS delivered. The EKLIPSE network particularly emphasizes some "physical indicators" related to the assessment of urban temperatures reduction, underlying the thermal regulation service of vegetated spaces. This is supported by the thickness of the edges connecting "indicators" to the "temperature reduction" and "tropical nights" nodes (Table A2). The multi-scale framework in which these indicators are computed also has to be underlined. Various scales are mentioned ("building", "street", "neighborhood", "mesoscale", "regional metropolitan", and so on) illustrating that NBS can act at different levels while they are implemented at the local scale. Thus, thermal comfort, energy use reduction and $CO_2$ sequestration/storage become significant in large areas (mesoscale) despite acting at the site/block scale (building) [4].

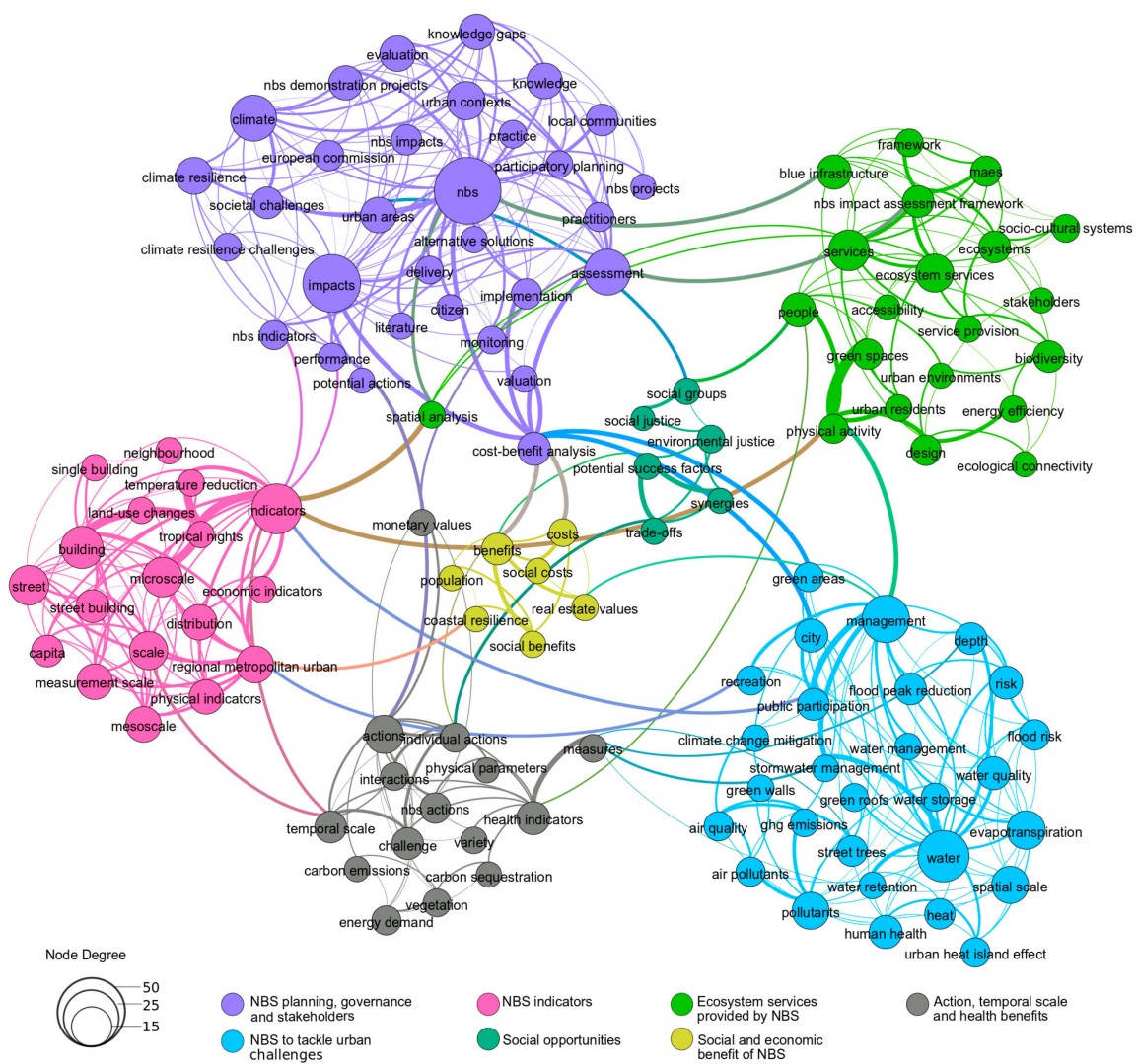

**Figure 2.** The EKLIPSE Network, co-occurrence of terms by conditional distance.

The next cluster, characterized by fewer nodes and accumulated degrees, corresponds to the "*Ecosystem services provided by NBS*", where nodes refer to ESS and additional intangible benefits like "accessibility", "recreation", "green spaces" and "blue infrastructure". Contrary to the quantitative

evaluation of the physical impacts by means of indicators in the previous cluster, the qualitative feature of these cultural services is dominant in this case. This cluster also contains the strongest connection of the whole network, between "physical health" and "green spaces" (Table A2). In this sense, Faivre et al. [16] pointed out that access to recreational green spaces may help to prevent socioeconomic inequality by promoting physical activity and public health. Likewise, it has been stressed that low quality of green spaces and public services in poor socio-economic neighborhoods make the population more vulnerable to the negative health impacts of extreme heat [26].

The remaining clusters are less remarkable due to a reduced number of nodes and lower node degrees. This is caused by a lower term occurrence along the text, leading to nodes with fewer connections in the network. However, we have attributed a name to each cluster; they correspond to "*Actions, temporal scale and health benefits*", "*Social benefit and economic opportunities of NBS*" and finally, "*Social opportunities*" for deprived groups and environmental justice addressed by urban green spaces. This implies that these topics have been studied less, or their impact is less known by the scientific community.

### 3.2. MAES-Urban Ecosystems

In MAES-Urban ecosystems, five nodes have degrees higher than 25 (Table A4). In order to assess the ESS in urban areas (the aim of MAES), "urban" and "ecosystems" stand out as the most important nodes of the network (Figure 3) with 35 and 34 degrees, respectively. Then, the distribution of "green landscape" (33) appears with the third highest degree of centrality, followed by "services" (30) and "urban ecosystems" (28).

Six clusters summarized the MAES-Urban Ecosystem report content (Table 4). The most important cluster is called "*Urban policies and NBS*", with 47 nodes and 420 accumulated node degrees. This cluster highlights the link between NBS and the way to integrate them in a sustainable urban planning policy. The highest computed node degrees ("urban ecosystems", "ecosystems", "nature-based solutions", "city council", "policy" or "local policymaking") strengthen this statement. In this context, the city council appears not only as a legitimated stakeholder that conciliates urbanization and urban ecosystems by means of policy at the city scale. It also has a protection and management role for the environment and the biodiversity, as demonstrated by the two strongest edges in the whole network (Table A4). According to Zwierzchowska et al. [24], urban policy documents are particularly relevant for the practical implementation of NBS. These define and organize the development of a city through the identification of key urban challenges and required actions to implement, in order to improve both environment and life quality. Furthermore, the European scope of MAES and its impact on local urban policies is well-stressed by nodes like "European commission", "European environment agency" or "member states".

**Table 4.** Clusters of MAES-Urban ecosystems, number of nodes and accumulated degrees.

| Cluster | Color | Number of Nodes | Accumulated Degree |
|---|---|---|---|
| Urban policies and NBS | Blue | 47 | 420 |
| Ecosystem services provided by NBS | Brown | 36 | 266 |
| Urban land use | Pastel Blue | 34 | 260 |
| Urban planning | Green | 28 | 117 |
| Ecosystem services assessment | Pink | 19 | 97 |
| Tools to assess ES | Yellow | 18 | 92 |

The "*Ecosystem services provided by NBS*" cluster appears to be the second most important by its accumulated degree (266) in the MAES report. Many nodes refer to different ESS: provisioning (e.g., "food", "crop fields", "community gardening", "urban allotments"), regulatory (e.g., "climate and temperature regulation", "water flow regulation", "flood protection", "noise", "pollination") and cultural services.

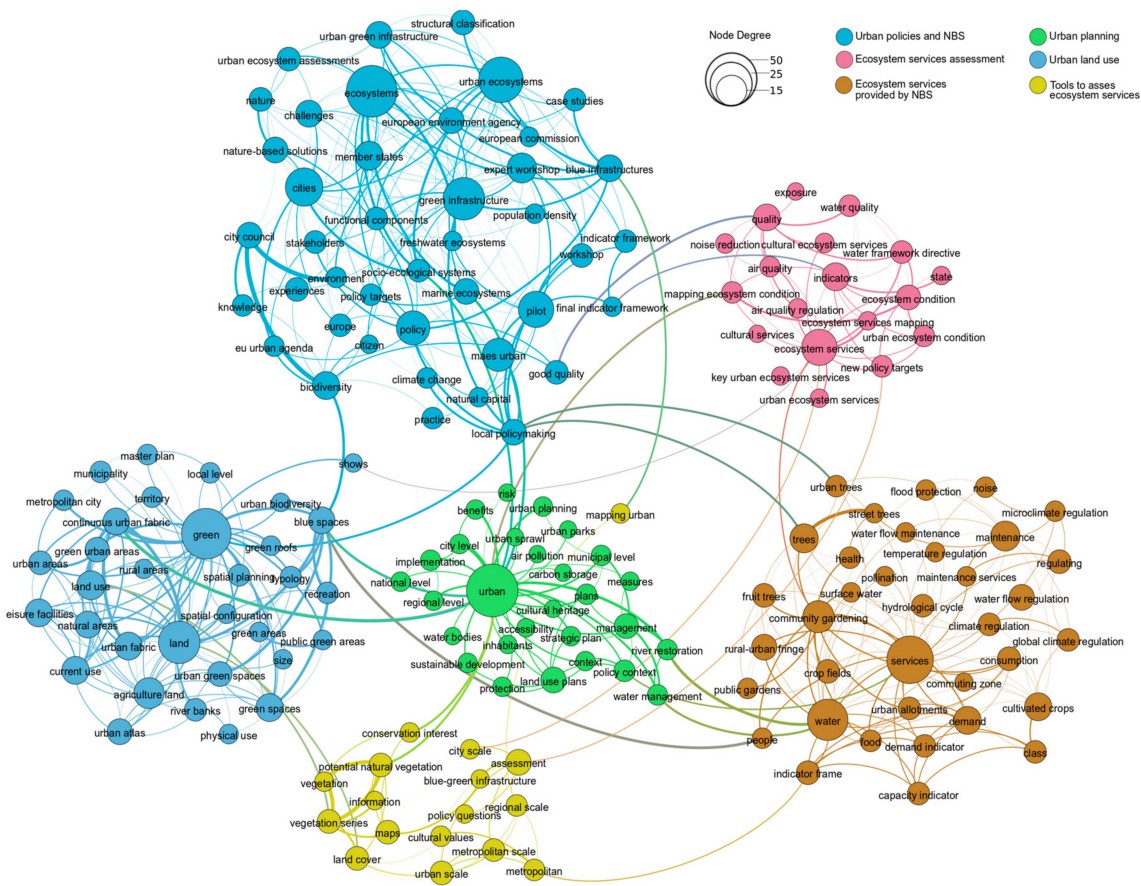

**Figure 3.** MAES-Urban Ecosystem Network, co-occurrence of terms by conditional distance.

The cluster "*Urban land use*" only presents the typology terms such as "green", "land", "agriculture land", "land use" or "green spaces" without a distinction of NBS type, except for "green roofs" and "riverbanks".

ESS and NBS benefits are integrated in the territory management at different administrative spatial scales in the "*Urban planning*" cluster, where terms like "urban", "land use plans", "management", "policy context", "implementation" or "plans" have a relevant node degree (Table A5). These nodes reveal some official planning instruments ("land use plans" or "spatial planning"), as a way to make operational the NBS policy and contribute to urban resilience [23].

In MAES, the need to assess the state of the urban ecosystems and their services via indicators is exposed through the "*Ecosystem services assessment*" cluster. Consequently, this cluster includes nodes like "ecosystem services" and "indicators" with a high centrality (high number of connections). In fact, this report developed two indicator frameworks based on the evidence of study cases to map urban green infrastructure, to assess their condition and to measure the urban ecosystem services delivered. Nodes such as "quality", "ecosystem condition" or "water framework directive" highlight the ecosystem assessment approach, while "air quality" or "noise reduction" nodes point to urban ecosystem services delivered. Finally, the tools to estimate these ESS at different spatial scales, such as "data", "maps" or "information", concern the cluster "*Tools to assess ESS.*"

*3.3. NATURVATION*

A noticeable characteristic of the NATURVATION network is its small size, with a reduced number of nodes (Figure 4). This is due to a shorter document length. Hence, there are fewer connections between terms than the other networks, as well as lower weight values (Table A6). Furthermore,

the most relevant nodes were extracted, finding 22 and 20 as the highest degrees of the network for "nbs" and "services" nodes, respectively.

In the NATure-based URban innoVATION report analysis, five clusters are identified (Table 5). "*Economic valuation*" represents the main cluster with 21 nodes and 137 accumulated degrees. The nodes with the highest degrees within this cluster correspond to "nature" (18), "valuation" (18) and "monetization" (12) (Table A7), revealing the economic approach chosen to assess the nature.

NBS and their impacts can be reflected as monetary benefits to be used as a support tool in public decision-making process. In case of area changes in open spaces, Brander and Koetse [22] pointed out that the estimated value per hectare of urban open space can be multiplied by the proposed changing hectares under an urban scenario policy, to provide the total annual value associated with the change. Such information can be used to decide whether to increase open space areas or to keep the existing ones.

In parallel, the financial benefits and monetization of green buildings can also support private investments [1]. Indeed, the monetary impact of NBS recreational, aesthetic or social benefits on the housing prices has also been demonstrated [27–30]. In fact, in the "*Economic valuation*" cluster appears the "housing prices" node close to "hedonic pricing". The former refers to an economic valuation method widely used to analyze the impact of UGI proximity (and their related amenities) on property values. The application of this method has displayed a different kind of recognition to NBS, in functions of their type. For example, there is a significant positive effect on housing prices for those facing water structures [27] or those close to forests and large parks. However, there is a negative impact on apartment prices surrounding cemeteries [28].

**Table 5.** Clusters of NATURVATION, number of nodes and accumulated degrees.

| Cluster | Color | Number of Nodes | Accumulated Degree |
|---|---|---|---|
| Economic valuation | Pink | 21 | 137 |
| Types of NBS | Blue | 19 | 110 |
| Ecosystem services provided by NBS | Green | 21 | 109 |
| NBS terms | Orange | 12 | 91 |
| Economic values of nature | Dark green | 11 | 54 |

The second most important cluster with 19 nodes emphasizes the variety of "*Types of NBS*" and refers to urban "spaces", including "green spaces" and "blue areas" nodes. Additionally, the "services" node is highlighted by the highest degree within the cluster, illustrating the importance of recognizing ESS. In this sense, the significant thickness of edges linking UHI effect with "services", "mitigation" and "recreation" (Table A6) reveals the need to promote NBS to improve thermal comfort perception and reduce UHI effects, but also to preserve the physical well-being of inhabitants, as indicated by C. O'Malley et al. [29].

The third cluster, "*Ecosystem Services provided by NBS*", extends in detail a set of ESS. It contains nodes like "local climate regulation", "air quality" or "carbon storage". All of them are recognized in the literature as regulating services, widely claimed, accepted and scientifically quantifiable [30]. Surprisingly, regulatory services related to water (stormwater management and flooding reduction) were not revealed by the network, meaning the node degrees were not high enough to be conserved.

Provisioning and cultural services are also present in this cluster; however, they are not widely detailed like the regulating services. Moreover, the "habitat" node appears close to "nbs", with a higher degree than "provisioning services". Indeed, NATURVATION methodology is used the ESS classification from the Millennium Ecosystem Assessment (MEA) [25], which includes habitat as a "supporting" service in the urban ecosystem.

NATURVATION follows the environmental economic valuation framework proposed by Hein et al. [31]. Then, four kinds of values are attributed to nature depending on their ESS usability principle: direct use value, indirect use value, option value and non-use value. These categories stand out in the

"*Economic Values of Nature*" cluster, which guides the economic assessment. This cluster does not have a direct connection with "*ESS provided by NBS*" cluster. Instead, these are linked through the "*Economic valuation*" cluster. This means that the monetary value of an ESS delivered by NBS can be assessed by an economic valuation approach as the "preference method", depending on the value that people can gain from ESS usability.

The "*NBS terms*" cluster presents nodes supporting the NBS concept, such as "green engineering", "blue infrastructure", "ecosystem approach", etc. This cluster is characterized by the highest homogeneity, with nodes degrees ranging from 7 ("blue infrastructure") up to 11 ("landscape domains"), as seen in Table A7.

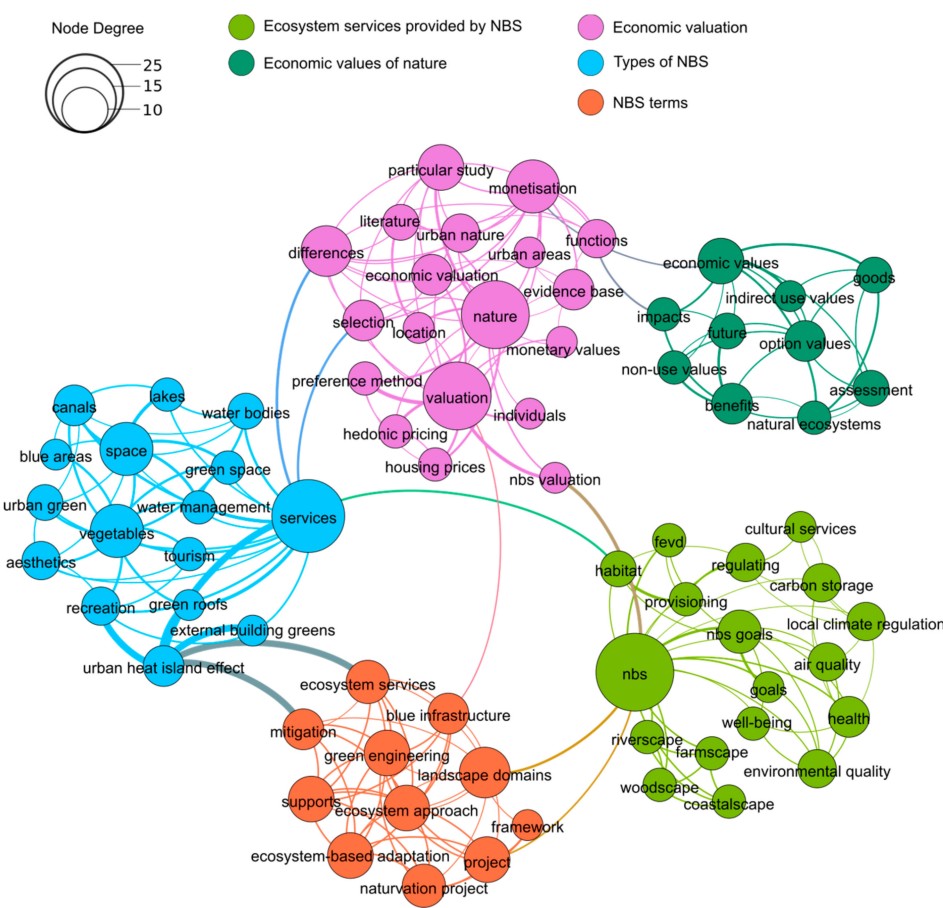

**Figure 4.** NATURVATION Network, co-occurrence of terms by conditional distance.

## 4. Discussion

From the individual analysis of the networks, it was possible to identify the main approaches and synergies of each document. Thank to this, a comparison of the results allows to continue developing hypotheses. Just like in the Results section, these are supported by the literature review.

Before making a cross comparison of these results, it must be recalled that the statistical properties of the corpora (paragraphs ang pages length, Table 2) determine the quality of the text-mining analysis (number of terms co-occurrence and clustering). The larger the corpus is, the more statistically relevant are the extracted outcomes. Consequently, the MAES—Urban Ecosystems and EKLIPSE networks propose consistent approaches with well-defined clusters, since they have a higher modularity. They have dense connections between nodes within clusters, but sparse connections between nodes belonging to different clusters. Concerning the NATURVATION network, the modularity decreases even if there are sparse connections between clusters. The connections within clusters are fewer and the nodes' degrees are more uniform.

It is important to note that some network connections may be missed or be false. In this study, the missing connections were easily detectable because of the length of the documents, which were analyzed in detail by the authors before exploiting them with text-mining software.

Nevertheless, this slight heterogeneity in the corpora does not seems to affect the results of this analysis, as they revealed some common advantages of NBS, as well as some differences (Table 6). Both are discussed below.

### 4.1. Ecosystem Services

Firstly, the three reports highlight different types of ESS delivered by NBS, and do not use the same framework to analyze them. EKLIPSE and NATURVATION both have a common ESS classification based on MEA [25], differentiating provisioning, regulating, supporting and cultural services. In the case of MAES—Urban ecosystem, the Common International Classification of Ecosystem Services (CICES) framework from the European Environment Agency (EEA) is used, which does not cover the supporting services defined in MEA.

In details, the regulating services provided by green spaces (e.g., climate regulation, water management, air quality regulation, etc.) appear at a higher rate of occurrence in the EKLIPSE and NATURVATION networks. These services are accented as a strategy of urban resilience, to mitigate environmental impacts resulting from urban sprawl and climate change effects. Usually, regulating services concerning climate mitigation and stormwater management are put forward by the scientific literature as the services predominantly appreciated in urban areas [20,30]. In consequence, the lack of reference to stormwater and flood management services is particularly surprising in the NATURVATION network, even if this was declared in the original corpus text.

Regarding the MAES—Urban ecosystems report, the full range of ESS is highlighted to tackle urban challenges. For example, the cultural services are well-identified, showing the benefits that the urban population gains from access to green and blue spaces: recreation, physical health promotion and socioeconomic inequality prevention.

### 4.2. NBS Assessment

The assessment of NBS efficiency to face urban challenges through indicators is well-exposed in two of the corpora but with different approaches. While EKLIPSE stresses the assessment of the delivered services (e.g., temperature reduction) and the multi-scale characteristic of their spatial impacts (e.g., building, neighborhood, microscale, regional), in MAES, two indicators frameworks are proposed: first to assess the urban ecosystem (e.g., state and condition) and then to measure the ESS delivered. Additionally, in MAES, the spatial scale is rather focused on the largest ones (e.g., regional, metropolitan and urban scale) to assess the impacts of a wide implementation of NBS (and their associated ESS). It is also important to highlight the preference for quantitative or physical indicators in EKLIPSE and MAES reports, to assess regulatory services. They are widely used and studied on a small scale (e.g., street, building), whereas qualitative assessment is usually chosen for cultural services.

Conversely, in the NATURVATION network, the term "indicators" is not put forward, since its occurrence in the corpus was too low to be considered by the statistical analysis. Nonetheless, the notion of assessment is present in the *Economic Values of Nature* cluster, pointing out a monetary value of the ESS delivered by NBS based on its usability.

Thus, the three corpora highlight NBS assessment as a useful mean to trace NBS effectiveness regarding a targeted urban challenge. As stressed by N. Kabisch et al. [20], this assessment becomes an useful support tool, which facilitates public and private decision-making processes on NBS investment and maintenance of sustainable urbanization.

**Table 6.** Main NBS topics, revealed by text mining and data visualization.

| Topic | EKLIPSE | MAES | NATURVATION |
|---|---|---|---|
| ESS classification | MEA, 2003 | CICES | MEA, 2003 |
| ESS highlighted | Regulating services<br>Cultural services | Regulating services<br>Provisioning services<br>Cultural services | Regulating services<br>Provisioning services<br>Cultural services<br>Habitat and supporting services |
| Assessment approach of NBS benefits | Indicators and multi-scale | Quality and condition of urban ecosystem | Assessment of nature value |
| Spatial scale approach and stakeholders of NBS deployment | Europe: EC<br>Mesoscale: Local communities, citizen | Europe: EC, EEA, European states.<br>Mesoscale: City council | Local: landscape domains. |

### 4.3. Multi-Scale Framework

As stated above, in the EKLIPSE project, the indicators proposed to assess NBS and their associated spatial scopes are studied in a multi-scale framework. However, a similar multi-scale approach is also observed in the networks concerning the stakeholders responsible for NBS deployment. EKLIPSE and MAES—Urban ecosystems emphasize a macroscale with a key role of the EU in NBS promotion. Indeed, different scientific studies argue that EC aims to position Europe as a world leader in global sustainability development by means of NBS and innovative sustainable solutions [16,24]. Text-mining analysis allowed to recognize the stakeholder role at the mesoscale (metropolitan or city level). At this scale, NBS implementation is performed by the local authorities through urban planning policies. In fact, MAES emphases the legitimated role of city councils to promote the goals and actions plans of NBS implementation, while EKLIPSE insists on the role of citizens in the decision-making process by the development of participatory initiatives. Various studies focusing on NBS governance highlight that NBS creation, design and implementation are successfully achieved when they involve multiple actors, mainly the citizens ensuring NBS recognition, acceptance and appreciation in the long term [20,32]. Regarding NATURVATION, no stakeholder is particularly associated with a spatial scale. However, it mentions different landscape domains where all kinds of UGIs are framed.

### 5. Conclusions and Perspectives

In this study, the main issues covered by three literature review EC projects reports of NBS impacts, benefits and trade-offs were explored and compared. Text-mining and visual data representation techniques were implemented to create networks based on the occurrence and co-occurrence of terms in the documents (corpora). Then, through the analysis of networks and their clusters, several hypotheses were made and supported by a conventional review of scientific literature.

Among the main NBS topics detected by the three documents, we found the large set of benefits offered by NBS in urban areas and their recognition as ESS. Equally important, the results enabled us to discover the importance of defining assessment tools of the benefits provided, such as performance, quality, or value indicators. Finally, some spatial scale and stakeholders involved in NBS implementation and maintenance were identified.

The results of this methodology are very valuable taking into consideration the variety of approaches and opinions of stakeholders involved in NBS deployment: EC, local communities and authorities, organizations, studies, etc. Particularly, the data visual representation of each corpus becomes a means or tool which facilitates communication between stakeholders.

The results are limited mainly on the length of the documents and the authors' approach. To get more accurate results in the future and a variety of opinions, this methodology could be applied to more documents, as well as to a set of documents in such a way as to create a large corpus. This will improve the statistical properties of the networks and the hypotheses supported.

In parallel, the NBS issues addressed in these reports of literature reviews demonstrated the understudied NBS horizons, which opens new perspectives concerning NBS impacts and benefits assessment. Regarding the convergence and the complementarity of the studied corpora, it appears relevant to define some new tools to assess NBS performances, not at a specific scale, but through scales. Moreover, such required tools should perform in an integrated way to consider every service provided by NBS (provisioning, regulating, supporting and cultural). It should be helpful for stakeholders to produce optimized NBS deployment scenarios depending on the targeted benefit(s) by estimating their potential impacts. Some initial works were carried out in this direction by conducting a fractal analysis on green roof implementation, to evaluate the relevance of the promoting policies [33]. Indeed, fractal analysis can be an innovative multi-scale approach for this purpose. In addition, other works are currently underway on the quantification of the ESS provided by NBS across scales. Based on multi-fractal theory, the temporal and spatial scales that characterize NBS performance for microclimate regulation are investigated. All of this is done through the characterization of ET measurements at the local scale and the analysis of green space distribution across different spatial scales and the socioeconomic dynamics of urban spaces (e.g., land use scenarios, population density, pressure of urban development).

**Author Contributions:** Conceptualization and methodology, L.A.C., I.T., P.-A.V., O.B.; Writing—original draft preparation, L.A.C.; Writing—review & editing, P.-A.V., O.B., I.T.; Supervision, P.-A.V., O.B., I.T. All authors have read and agreed to the published version of the manuscript.

**Funding:** This research is developed in the framework of the ANR EVNATURB project aiming to evaluate the ecosystem performances for renaturing urban environments. Additionally, partial financial support of the Chair "Hydrology for Resilient Cities" endowed by Veolia group is gratefully acknowledged.

**Acknowledgments:** The authors are very grateful for the valuable technical support given by the CNRS Complex Systems Institute of Paris Île-de-France (ISC-PIF) and the fruitful comments and suggestions given by Fréderique Bordignon from École des Ponts ParisTech.

**Conflicts of Interest:** The authors declare no conflict of interest.

## Appendix A

**Table A1.** General NBS projects information.

| Project | Start Date | End Date | Budget | Framework | Coordinator |
|---|---|---|---|---|---|
| EKLIPSE | 1 February 2016 | 31 July 2020 | € 2,997,272.49 | H2020-EU | United Kingdom Research and Innovation |
| NATURVATION | 1 November 2016 | 31 October 2020 | € 7,797,877.50 | H2020-EU | University of Durham |
| ESMERALDA | 1 February 2015 | 31 July 2018 | € 3,133,306 | H2020-EU | Gottfried Wilhelm Leibniz Universitaet Hannover |
| MAES—Urban Ecosystems | | | | Action 5 of the EU Biodiversity Strategy to 2020 | - Joint Research Center <br> - Dutch National Institute for Public Health and the Environment (RIVM) <br> - European member states |

**Table A2.** Main attributes of the EKLIPSE network.

| Nodes Degree | | Edge Weight | |
|---|---|---|---|
| *Label* | *Degree* | *Label* | *Weight* |
| NBS | 36 | Physical activity-Green spaces | 0.182 |
| Impacts | 29 | Building-Tropical nights | 0.143 |
| Indicators | 22 | Tropical nights-Indicators | 0.143 |
| Water | 22 | Tropical nights-Temperature reduction | 0.143 |
| Climate | 19 | Tropical nights-Microscale | 0.143 |

**Table A3.** List of main nodes EKLIPSE by cluster. (**a**) NBS Planning, Governance and Management, (**b**) NBS to Tackle Urban Challenges, (**c**) NBS Indicators and Scale, (**d**) Ecosystem Services Provided by the NBS, (**e**) Action, Temporal Scale and Health Benefits, (**f**) Social Benefit and Economic Opportunities if NBS and (**g**) Social Opportunities.

| (a) | | (b) | |
|---|---|---|---|
| **NBS Planning, Governance and Management** | | **Ecosystem Services Provided by the NBS** | |
| *Node* | *Degree* | *Node* | *Degree* |
| nbs | 36 | services | 16 |
| impacts | 29 | ecosystem services | 13 |
| climate | 19 | blue infrastructure | 12 |
| assessment | 19 | biodiversity | 10 |
| urban contexts | 13 | people | 10 |
| **(c)** | | **(d)** | |
| **NBS Indicators and Scale** | | **Ecosystem Services Provided by the NBS** | |
| *Node* | *Degree* | *Node* | *Degree* |
| indicators | 22 | services | 16 |
| building | 16 | ecosystem services | 13 |
| street | 14 | blue infrastructure | 12 |
| microscale | 14 | biodiversity | 10 |
| scale | 13 | people | 10 |
| **(e)** | | **(f)** | |
| **Action, Temporal Scale and Health Benefits** | | **Social Benefit and Economic Opportunities of NBS** | |
| *Node* | *Degree* | *Node* | *Degree* |
| actions | 13 | benefits | 12 |
| health indicators | 9 | costs | 5 |
| challenge | 8 | real estate values | 4 |
| temporal scale | 8 | social benefits | 4 |
| nbs actions | 7 | coastal resilience | 3 |
| **(g)** | | | |
| **Social Opportunities** | | | |
| *Node* | *Degree* | | |
| environmental justice | 5 | | |
| social groups | 5 | | |
| trade-offs | 4 | | |
| potential success factors | 4 | | |
| synergies | 3 | | |

**Table A4.** Main attributes of MAES—Urban ecosystems network.

| Node Degree | | Edge Weight | |
|---|---|---|---|
| *Label* | *Degree* | *Label* | *Weight* |
| Urban | 35 | Environment-City council | 0.097 |
| Ecosystems | 34 | Biodiversity-City council | 0.097 |
| Green | 33 | City level-Urban | 0.093 |
| Services | 30 | Vegetation-Vegetation series | 0.088 |
| Urban ecosystems | 28 | Potential natural vegetation-Vegetation series | 0.088 |

**Table A5.** List of main nodes MAES—Urban Ecosystem by cluster. (**a**) Urban Policies and NBS, (**b**) Ecosystems Services Provided by NBS, (**c**) Urban Land Use, (**d**) Urban Planning, (**e**) Ecosystem Services Assessment and (**f**) Tools to Assess ESS.

| (a) | | (b) | |
| --- | --- | --- | --- |
| **Urban Policies and NBS** | | **Ecosystem Services Provided by NBS** | |
| *Node* | *Degree* | *Node* | *Degree* |
| ecosystems | 34 | services | 30 |
| urban ecosystems | 28 | water | 24 |
| green infrastructure | 25 | community gardening | 15 |
| cities | 21 | maintenance | 13 |
| pilot | 19 | trees | 12 |
| **(c)** | | **(d)** | |
| **Urban Land Use** | | **Urban Planning** | |
| *Node* | *Degree* | *Node* | *Degree* |
| green | 33 | urban | 35 |
| land | 25 | land use plans | 9 |
| agriculture land | 14 | management | 8 |
| land use | 12 | policy context | 5 |
| green spaces | 11 | implementation | 4 |
| **(e)** | | **(f)** | |
| **Ecosystem Services Assessment** | | **Tools to Assess ESS** | |
| *Node* | *Degree* | *Node* | *Degree* |
| ecosystem services | 19 | data | 10 |
| indicators | 11 | assessment | 9 |
| quality | 10 | maps | 7 |
| ecosystem condition | 8 | land cover | 7 |
| water framework directive | 6 | urban scale | 7 |

**Table A6.** Main attributes of the NATURVATION network.

| Node Degree | | Edge Weight | |
| --- | --- | --- | --- |
| *Label* | *Degree* | *Label* | *Weight* |
| NBS | 22 | UHI effect- Services | 0.143 |
| Services | 20 | UHI effect- Ecosystem services | 0.143 |
| Nature | 18 | UHI effect- Mitigation | 0.143 |
| Valuation | 18 | UHI effect- Recreation | 0.143 |
| Space | 12 | UHI effect- Green roofs | 0.143 |

**Table A7.** List of main nodes of NATURVATION by cluster. (**a**) Economic Valuation, (**b**) Types of NBS, (**c**) Ecosystem Services Provided by NBS, (**d**) NBS Terms and (**e**) Types of Values.

| (a) | | (b) | |
| --- | --- | --- | --- |
| **Economic Valuation** | | **Types of NbS** | |
| *Node* | *Degree* | *Node* | *Degree* |
| nature | 18 | services | 20 |
| valuation | 18 | space | 12 |
| monetization | 12 | vegetables | 12 |
| differences | 11 | canals | 7 |
| particular study | 9 | urban heat island effect | 7 |

**Table A7.** *Cont.*

| (c) | | (d) | |
|---|---|---|---|
| **Ecosystem Services Provided by NBS** | | **NBS Terms** | |
| *Node* | *Degree* | *Node* | *Degree* |
| nbs | 22 | landscape domains | 11 |
| nbs goals | 8 | project | 9 |
| health | 7 | supports | 9 |
| regulating | 6 | ecosystem approach | 9 |
| air quality | 6 | ecosystem-based adaptation | 9 |

| (e) | |
|---|---|
| **Types of Values** | |
| *Node* | *Degree* |
| economic values | 9 |
| benefits | 7 |
| option values | 7 |
| assessment | 5 |
| goods | 5 |

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
