# Peer review of "A Text-Mining Approach to Compare Impacts and Benefits of Nature-Based Solutions in Europe"

_sustainability, doi:10.3390/su12187799_

Round 1

Reviewer 1 Report

The manuscript entitled “A Text-Mining Approach to Compare Impacts and Benefits of Nature-Based Solutions in Europe”.  The article uses a text mining analysis to convert the reports into a meaningful structure. The selected reports focus mostly on NBS at the urban scale. The visualization of the results of the article is very interesting (e.g., the clusters of each report).  Accordingly, I do think that this article should be considered for publication. However, I have major doubts if the presented article really helps analyses the impacts and benefits of NBS. My main concerns are the following:

  1. Introduction: i) you have mentioned several terms that connect to NBS concept. However, a few terms used in Europe are still missing e.g., Sustainable Urban Drainage Systems (SuDS), and Ecosystem-based Adaptation (EbA). It would help if you discussed how these terms similar or different from NBS. ii) Line 61. You mentioned that “the EC has funded several projects to carry out an extensive review of the available literature about their physical, social and economic impact.” I am not sure is this state is correct that the EC has funded several projects to carry out an extensive review but I think EU has funded several projects to develop a framework, an application guide for NBS, platform, etc. Many projects have used extensive review as one of the methods for their projects. iii) Line 73, you mentioned that “allowing to recognize concepts as well as to identify stakeholders and their position”. This sentence is not clear to me. Can you explain it in more detail?
  2. Methodology i) Please check the sentence line 104. If I understand correctly, the challenges are not the results from the review process, but the review is used to develop NBS Impact Assessment Framework. Based on the report, 10 challenges are selected by EWG from the expert report on NBS supported by DG Research and Innovation (European Commission, 2016) and a recent review of NBS frameworks (Kabisch et al., 2016). ii) Please put the references for the report line 108 or in Table 1. iii) Text mining analysis may be more useful for a larger analysis, but in your research, you selected only 3 reports, which is possible to read, and we will get information by reading. What is the benefit of text mining in this case? iv) There is nothing new in term of the methodology as you used the exiting software to analyze the research. Therefore, I think this should be mentioned clearly in discussion or Introduction that your contribution is not on developing the text-mining approach, but on the analysis. v) The methodology of the text-mining analysis is too general. You should mention a more specific setting that you have done for you research. For example, line 160 “terms-status modification and a lexicographic analysis can be made by the user”. In this case, you can give an example of how did you do it.
  3. Results i) I understand that you have some references for the statement “These terms show the need to reconsider the urban planning process, encouraging NBS in urban areas, as a climate resilience strategy to create urban sustainability as argued in the recent literature [13,15,17]”. Based, on the your results, this could mean that to do the urban planning process; these terms often used to in NBS planning, governance and stakeholders. ii) Could you comment or discuss on why the number of nodes of “Actions, temporal scale and health benefits”, “Social benefit and economic opportunities of NBS” and “Social opportunities” (line 257) are small.
  4. Discussion & Perspectives i) Can you be more specific on what is the need (line 423) “Thus, the three corpuses highlight the need to assess NBS effectiveness/impact” ii) a more in-depth discuss on new tools to assess NBS performances line 447. iii) Findings made should be put more profoundly into the context of the relevant body of knowledge. The methodology itself should also be evaluated more thoroughly, as there is no discussion on the limitations, advantages, or disadvantages of the text-mining approach. iv) Discussion on how this research will benefit other research or how the research contributes to the sciences.

Author Response

We would like to thank you for your constructive comments and suggestions, very useful to improve the paper.

Reviewer 2 Report

I find the paper interesting and well-written. However, I have some concerns related to the results and the conclusions supported by the research. Text-mining process application is adequate but some results look not supported by this process. The identification of the results, supported by the cluster analysis, and the related conclusions, based on a critical analysis of the projects, need to be improved. In this sense, the result section should be reviewed and a conclusion section should be included, identifying specific research contributions.

  1. Line 51-52: In the manuscript, the following sentence is referred “Further, the implementation of different NBS facilities - such as green roofs, rain gardens, green spaces, green swales, permeable pavements and retention basins - proved the minimization of both, peak discharge and runoff volumes by up to 90% on all the range of rainfall event’s return periods at the urban project scale”. Is this statement a conclusion of the reference [8]? 90% represents a really high reduction. The water retention and other related hydrological processes depend on several conditions such as the previous dry weather, the maintenance conditions, among others. I recommend to refer these other factors.
  2. Line 54: urban microclimate? I recommend to use the more specific term “Urban Heat Island”. Indeed, related so urban parks, this term is referred posteriorly by the authors.
  3. Line 91-92: The aim of this research is compare the results of three projects. This analysis is carried out based on three reports? Why the research does not include more reports of each project? This seems like not sufficiently representative. An extensive and consolidate statistical process is applied, however, each project is characterized based only in one report. Can you explain why only three reports were selected?
  4. Table 2. I think that table 2 is not required. This information can be detailed in a sentence.
  5. I believe that the incorporation of a scheme to describe the text-mining process and the result visualization can be useful.
  6. I have some concerns and difficulties to identify the results presented. If I understood correctly, the research results are based on the clusters analysis. It is correct? In essence, several words are aggregated in a cluster according to a specific rule/distance. The results are also based on a critical analysis of the 3 reports carried out by the authors? Please, review carefully the results section and identify clearly which results are supported by the cluster analysis.

For example, in line 201, how it possible to identify the following conclusion: “This classification reveals the need to face environmental and urban challenges by means of NBS and the way to assess their efficiency”. Why a need? This conclusion is based on the NBS, impacts and indicators degrees? Please, include a better explanation regarding how this result was identified. This is just an example. There are several conclusions along the result section that need a better explanation.

  1. Line 375: please, give an example of the statistical properties referred.
  2. Table 6: NBS characteristics? I believe that the information presented in this table corresponds to NBS concerns or NBS topics analysed in these projects.
  3. Include a specific conclusion sections, identifying clearly the contributions based on this research.

Author Response

We are grateful for your comments, which were of huge help to improve and adopt the results to the paper object.

Round 2

Reviewer 2 Report

The authors have improved the manuscript, taking into consideation the comments provided. 

I agree with the authors' modifications and their answers.